# Identification of QTLs for Spot Blotch Resistance in Two Bi-Parental Mapping Populations of Wheat

**DOI:** 10.3390/plants10050973

**Published:** 2021-05-13

**Authors:** Navin C. Gahtyari, Chandan Roy, Xinyao He, Krishna K. Roy, Mohamed M. A. Reza, Md. A. Hakim, Paritosh K. Malaker, Arun K. Joshi, Pawan K. Singh

**Affiliations:** 1ICAR—Vivekanand Parvatiya Krishi Anushandhan Sansthan, Almora, Uttarakhand 263601, India; navin.gahtyari@icar.gov.in; 2Department of Plant Breeding and Genetics, Bihar Agricultural University, Sabour, Bihar 813210, India; chandan.roy43@gmail.com; 3International Maize and Wheat Improvement Center (CIMMYT), Apdo. Postal 6-641, 06600 Mexico DF, Mexico; x.he@cgiar.org; 4Bangladesh Wheat and Maize Research Institute (BWMRI), Nashipur, Dinajpur 5200, Bangladesh; rkrishnaroy666@gmail.com (K.K.R.); marezawrc@gmail.com (M.M.A.R.); hakimwrc@bari.gov.bd (M.A.H.); pkmalakerwrc@gmail.com (P.K.M.); 5CIMMYT/Borlaug Institute for South Asia, NASC Complex, DPS Marg, New Delhi 110012, India; a.k.joshi@cgiar.org

**Keywords:** *Triticum aestivum*, *Bipolaris sorokiniana*, QTLs, disease resistance, *Sb4*, *Lr46*

## Abstract

Spot blotch (SB) disease caused by the hemibiotrophic pathogen *Bipolaris sorokiniana* inflicting major losses to the wheat grown in warm and highly humid areas of the Indian subcontinent, including Bangladesh, necessitates identification of QTLs stably expressing in Indian subcontinent conditions. Thus, two RIL mapping populations, i.e., WC (WUYA × CIANO T79) and KC (KATH × CIANO T79), were phenotyped at Dinajpur, Bangladesh for three consecutive years (2013-2015) and genotyped on a DArTseq genotyping by sequencing (GBS) platform at CIMMYT, Mexico. In both populations, quantitative inheritance along with transgressive segregation for SB resistance was identified. The identified QTLs were mostly minor and were detected on 10 chromosomes, i.e., 1A, 1B, 2A, 2B, 2D, 4B, 4D, 5A, 5D, and 7B. The phenotypic variation explained by the identified QTLs ranged from 2.3–15.0%, whereby QTLs on 4B (13.7%) and 5D (15.0%) were the largest in effect. The identified QTLs upon stacking showed an additive effect in lowering the SB score in both populations. The probable presence of newly identified *Sb4* and durable resistance gene *Lr46* in the identified QTL regions indicates the importance of these genes in breeding for SB resistance in Bangladesh and the whole of South Asia.

## 1. Introduction

Wheat (*Triticum aestivum* L.) grown in the warm and humid regions of the world, in particular South Asia, is vulnerable to many biotic stresses, among which spot blotch (SB) caused by *Cochliobolus sativus* (anamorph *Bipolaris sorokiniana*, syn. *Helminthosporium sativum*) is a significant problem [1,2]. The disease is estimated to cause 15–20% average yield loss in South Asia [3], however, under favourable conditions, more than 85% losses during the summer season in Zambia [4] and on susceptible wheat cultivars [5] were observed. Specific studies in Bangladesh also indicate as much as 60% of the crop was affected with yield losses to the tune of 22% [6]. Typical symptoms that are visible on leaves, sheath, and glumes are light brown coloured oblong to elliptical lesions, which gradually coalesce to increase in size and become necrotic [7]. In severe cases, the grains in the infected spikes become shrivelled with a characteristic black point towards the embryo end [8]. This pathogen is known to cause multiple diseases in wheat such as seedling blight, seedling rot, common root rot and seed rot [9]. The disease mostly occurs singly, but in some regions, it may occur as a complex of SB and tan spot, better known as Helminthosporium leaf blight (HLB), where the former is found to be more predominant and infective [1,10]. To fight the SB menace, a special initiative by CIMMYT was undertaken in 2009 especially for disease-prone South Asian countries in the form of a special nursery called the CSISA-SB (Cereal System Initiative for South Asia—spot blotch). During the course of this initiative, it was found that, as suggested earlier [11], increased temperature and sporadic rainfalls might prompt SB to spread to non-traditional regions of the world [12]. Hence CSISA-SB was reconstituted into Helminthosporium leaf blight screening nursery (HLBSN) in 2015 with a broadened vision to supply agronomically superior breeding lines coupled with SB resistance to not only South Asia but other affected regions of Africa and Latin America [11,12].

Both seed treatments and foliar fungicidal sprays are advocated for use against SB disease. Seed treatments with a mixture of carboxin and thiram or fludioxonil and difenoconazole not only limit the disease severity but also improve seed germination in the fields [13,14]. Likewise foliar sprays with a number of fungicidal molecules like carbendazim, difenoconazole, propiconazole, and azoxistrobin have been tested against SB infection and found to reduce yield reduction to the tune of 10–30% [14]. The high efficacy of triazole fungicides like tebuconazole and propiconazole is attributed to their ability to limit SB infection by interfering with fungal cell wall synthesis [14,15]. Despite the proven efficacy of fungicides, they have not been widely applied due to high costs, limited availability, and the hazard to the environment. This is especially true for South Asian countries, where 80–90% of farmers are smallholders [3,9]. Thus, an integrated disease management approach with genetic resistance as the main component is advocated to achieve management of SB. An immune reaction against the SB disease has not been reported yet, however, a number of wheat genotypes and accessions with good levels of genetic resistance have been reported, among which some have been used as a resistant parent in QTL studies [7,11]. Several resistant sources have been identified in the Eastern Gangetic Plains (EGP) of South Asia (Bangladesh, India, Nepal), such as Chirya1, Chirya7, NL 623, NL 644, NL 297, Milan/Shanghai #7 etc. [16,17] and in other parts of the world—Yangmai6, CNT1, Shanghai #4, Suzhoe #8 [11]. Wild wheat relatives and alien species, *Aegilops squarrosa*, *Agropyron curvifolium* etc. are also reported to harbour SB resistance with special reference to *A. curvifolium* (syn. *Thinopyrum curvifolium*) since introgression from it resulted in the identification of several Chirya lines resistant to SB [11]. Many of these resistant genotypes were used as a parent for identifying underlying QTLs/genes for SB resistance, including ‘Ning 8201’, ‘Chirya3’ [18], ‘Saar’ [19], ‘YS116’, ‘Yangmai6’ [20], ‘BH 1146’ [21], a synthetic derivative ‘SYN1’ [22], winter wheat line 621-7-1 [23], ‘Zhongyu1211’, ‘GY17’ [24] etc.

Conventional genetic studies have identified both quantitative as well as qualitative inheritance for SB resistance. Additive, dominant, partially dominant and even recessive genes controlling SB resistance have been reported [25,26,27,28]. However, recent association/QTL studies indicated the quantitative inheritance for SB resistance [20,29,30,31], though four genes (*Sb1* to *Sb4*) where large effects on SB resistance have been identified. *Sb1* is located on chromosome 7DS co-localizing with *Lr34* [19], whereas *Sb2*, *Sb3* and the newly identified *Sb4* have been mapped on chromosomes 5BL [20], 3BS [23], and 4BL [24], respectively. The other reported QTLs for SB resistance are mostly minor in effect and found to be widely dispersed in the genome i.e., except for 3D and 5D, all wheat chromosomes were identified as having QTLs for SB resistance [31]. Since SB is the most important wheat disease in Bangladesh and the EGP of South Asia, it is important to detect QTLs effectively in this particular region. Though QTLs have been identified for SB resistance, many similar studies were conducted at the seedling stage in the greenhouse [32,33,34], which does not well reflect the field resistance. Additionally, field studies deciphering the QTLs and underlying effective genes against SB though available, are limited in number, and were also performed in a limited area with less diverse environmental conditions [18,19]. Therefore, two populations—WC (WUYA × CIANO T79) and KC (KATH × CIANO T79)—previously phenotyped in Mexico [30,31], were used for QTL mapping by phenotyping at a much farther intercontinental distance at Dinajpur, Bangladesh, a hot spot for the SB incidence, to obtain confirmation about QTLs operating in South Asia.

## 2. Results

### 2.1. Phenotypic Data

Analysis of variance for both populations indicated significant variation in disease pressure across the three years with genotypes significantly interacting with year (Table 1). A Levene’s test indicated non-significant differences between replications for both populations in 2013 and 2014, but not in 2015, thus the two replications of both populations in 2015 were analysed separately for QTL mapping.

Within the same year, both populations responded similarly as observed from their comparable disease severity/AUDPC score ranges, with the highest disease range observed in 2014 for both populations. For both WC and KC populations, a continuous distribution in SB scores over three years, with parental values lying in between the distribution, indicated the presence of transgressive segregation for both resistance and susceptibility; however, an exception was noted in 2013 and 2015 for the KC population where both resistant and susceptible parents scored similarly for disease severity (Figure 1). For the WC population, the resistant parent WUYA (181.53) was comparable to resistant check Chirya-3 (172.91) in average SB score. Interestingly, 74 RILs were found to transgress resistance with an average SB score of 170.36. For the KC population, the resistant parent KATH (241.46) was transgressed by 16 RILs for resistance with an average SB score of 210.04. However, only one RIL (158.02) was able to beat the resistant check Chirya-3 (161.94) in averaged SB scores for three years (Appendix A). Plant height (PH) and days to heading (DH) showed significant but negative correlation with SB scores in both populations, barring non-significant (though negative) results for DH (2014 and 2015) in the KC population. Heritability varied from 0.30 to 0.77 for WC and 0.68 to 0.91 for KC populations. PH showed a higher magnitude of negative correlation compared to DH in both populations (Table 2).

### 2.2. Linkage Map

The anchored markers helped to form 21 linkage groups representing 21 wheat chromosomes in the two mapping populations. A total genetic distance of 3792.4 cM and 3517.9 cM with a comparable average marker density of 1.5 and 1.6 cM between markers was mapped for KC and WC populations, respectively (Table 3).

The genome of WC was mapped with an average genetic distance of 180.6 cM/chromosome with chromosomes 4D and 5D having the least coverage (<100 cM), whereas for the KC population the average genetic distance was 167.5 cM/chromosome with chromosomes 1A and 2D having the least coverage (<100 cM). A and B genomes were more saturated (1.0–1.5 cM between markers) and covered a greater genetic distance than the D genome with an average marker density of 5.2 cM/marker. The markers on the D genome covered 26.6% (198 markers) and 27.1% (180 markers) genetic distance in the WC and KC population, respectively. A few gene-based markers such as *Rht-B1* and *Vrn-A1* were mapped in both populations.

### 2.3. QTL Mapping

#### 2.3.1. WC Population

Altogether 11 different QTLs located on 8 chromosomes were identified in the present investigation (Figure 2). The identified QTLs explained the highest cumulative phenotypic variation in the year 2013 (59.7%) followed by 2014 (42.0%) and 2015 (14.4%), with the mean percentage variation explained (PVE) of 44.6%. Eight QTLs on 1A (marker interval 1218247–1096735), 1B (1137809–5411162), 2A (3020873–1132194), 2A (4991898–3064660), 4B (985312–1241652), 4B (1195526–100167524), 5A (1102419–5411712) and 7B (7341261–1007925) were found in more than one year (Table 4).

The identified QTLs were mostly minor with phenotypic effects ranging from 2.4–15.0%, and appeared to act in an additive mode when stacked (Appendix A). Three QTLs exhibiting large phenotypic effects (PVE of 9.9 to 15.0%, mainly in the year 2013) were detected on chromosomes 1A, 4B and 5D, among which QTLs on 1A (82.5 cM) and 4B (40.5 cM) seemed to be somewhat stable (2013, 2014 and mean year). Six QTLs for PH and one for DH dispersed on 7 chromosomes were also detected. Among them, QTLs for PH on 1A, 2A and 4B and that for DH on 5A co-localized with SB resistance QTLs (Appendix A). When adjusted for PH and DH, the QTLs associated with these two traits exhibited reduced average PVE for SB resistance by 50.36% and 30.74%, respectively.

#### 2.3.2. KC Population

Eight QTLs spread over 4 chromosomes explaining phenotypic variations from 2.3 to 11.5% were identified (Figure 3, Table 4). From these, four QTLs—2B (1218621–4992694), 2B (1066619–1278607), 4B (3222467–1233562) and 4D (*Rht-D1*-BS00036421)—were detected for more than one year including the SB mean scores (Table 4). Like the WC population, the identified QTLs for the KC population were also minor (averaged 6.6% PVE), showing additive effects when stacked in a genotype (Appendix A). Nevertheless, two moderate-effect QTLs, one on chromosome 4B (3222467–1233562) with 10.8% and 11.5% PVE in 2013 and 2015, respectively, and the other on chromosome 4D (*Rht-D1*-BS00036421; 9.1% PVE) in 2015 were detected. Seven QTLs for phenological traits namely, three for PH and four for DH were detected on 6 chromosomes. Among them, QTLs on 2B for DH and on 4B, 4D for PH co-localized with those for SB resistance (Appendix A). When adjusted for PH and DH, the QTLs associated with the two traits exhibited reduced average PVE for SB by 43.66% and 30.51%, respectively.

#### 2.3.3. Bulk Studies

Ten extremely resistant and susceptible RILs in both populations were marked for each of the identified QTL (Table 5). When a favourable (resistant) allele for SB resistance is present for both left and right markers flanking the QTL, then only QTL is marked to be present. QTL at 94.2cM on chromosome 2A, 59.0 cM on 4B, and 72.5cM on 2D were in a higher frequency in the resistance bulk compared to the susceptible bulk, and thus were the major differing QTLs between the two bulks in the WC population. Similarly, QTLs on 2A (48.4cM), 2B (17.9cM) and 4D (15.7 cM) differed between the resistant and susceptible bulks in the KC population. The stacking of these DRSB (different between resistant and susceptible bulks) QTL in WC and KC was observed with lowered SB disease scores (Figure 4).

## 3. Discussion

Although wheat crops are vulnerable to SB in Bangladesh and other parts of the EGP in South Asia, they become much more vulnerable when sown late, which coincides with warmer temperatures and rain during the grain filling stages [35,36]. Therefore, the sowing of the experimental crop in this study was delayed deliberately to enhance SB pressure and to detect effective QTLs governing SB resistance. During 2014, the WC population skewed more toward the resistance side whereas an opposite trend was noted for the KC population. Differential skewness has been observed in past research and might be an indicator of the presence of different resistance genes in the two populations [27]. Nevertheless, for all three years and in both populations, a continuous distribution of RILs with transgression to both resistant and susceptible directions signified the quantitative mode of inheritance. Earlier studies also indicated polygenic control of SB resistance with many small-effect QTLs governing the trait [7,31].

Both exclusive (1A, 1B, 2D, 5D, 7B for WC; 2B, 4D for KC population) and common QTL sites (2A and 4B) as per position on the Chinese Spring reference genome were identified in the present study. Although lower genetic coverage was observed for the D genome compared to A and B genomes, the present study was able to locate SB resistant QTLs on 2D, 4D and 5D. Many past studies have also indicated low polymorphism in the D genome, which may be attributed to its recent introgression into the wheat genome [37,38]. QTLs on 1A, 1B, 1D, 5A, 7B, 7D had been identified in diverse association panels of spring wheat, landraces and in RILs of bi-parental crosses [21,29,32,39], where some of these QTLs (1A, 1B, 7B, 7D) decreased SB severity mainly by reducing the lesion number [29] and had been identified in the comparable environmental conditions of Coochbehar and Kalyani [21]. The 1A QTL in WC might represent a new QTL since it was positioned at 506.7 Mb, whereas the previously reported QTLs were located at around 44.03 Mb [39].

The identified QTL on 1B was repeatedly detected in the WC population. This QTL is the same as detected in a previous study in Mexico [30] but different from the one detected by Gurung et al. [34]. The projected location of this QTL on CS was within a 3.1 Mb region (670.6–673.7 Mb) which matched the position of *Lr46* [40] and is in close proximity with the earlier reported regions on 1B from 621.2 to 674.0 Mb [31,39]. Therefore, this QTL appears robust by the fact that it expressed stably across multi-environments in all four studies (including the present) [30,31,39]. *Lr46* had been previously associated with SB resistance along with *Lr34* [19] and its association with the morphological marker, leaf tip necrosis (LTN), makes it breeder friendly. In fact, Joshi et al. [41] demonstrated the association of LTN with lowered SB scores in >1400 wheat accessions and in segregating generations of crosses having contrasting parents for LTN. Bainsla et al. [39] associated this genomic region with an NBS-LRR gene family (TraesCS1B01G416200) coding for a disease resistance protein. Hence, the detected QTL in the present study might be similar in its mechanism to provide durable resistance against SB. 

In both populations, QTLs on the group 2 homeologous chromosomes differed significantly among the resistant and susceptible bulks. Among these DSRB QTLs, the 2A QTL on WC seems to be linked with SSR marker *Xgwm425* which had been successfully used to introgress SB resistance by marker-assisted backcrossing [42]. QTLs on 2B and 2D are similar to the ones reported in the Mexican conditions as deduced either by their physical positions or by the contribution of the SB resistance allele [30,31]. The latter QTL located at 63.74 Mb is quite far away from the previously located QTLs at 389.46 Mb [43] and 607–648 Mb [33,39], indicating that it is a novel QTL. Interestingly, all these DRSB QTLs as placed at the initial 90 Mb regions of the homeologous chromosomes were free from the confounding effects of DH or PH, indicative of their importance for imparting SB resistance. The fact becomes more affirmative from the recent work of Zhang et al. [24]. They pointed out that chromosome 2B has an important QTL (having five differential SNPs) after 4B (where they identified *Sb4* gene), and in the present study as well, it was detected somewhat stably in multi-environments. The appearance of these QTLs in multiple environments along with their association with resistant RILs in a higher frequency indicates probable underlying genes/mechanisms important for imparting resistance.

The negative association between SB severity and phenological traits PH and DH as detected in the present investigation agrees well with those reported previously in diverse environments [21,30]. Generally, late and taller varieties are found to be SB resistant, though this particular association is highly undesirable due to lodging and other biotic/abiotic stresses, especially in the era of climate change, particularly for the South Asian region. Thus, scoring these traits in field conditions and genotyping with *Rht* (4B, 4D) and *Vrn* (5A) genes as performed in the present study become important to identify undesirable linkage and to select useful segregants without any linkage drag. Both types of QTLs, i.e., confounded with height/heading effects and free from them, were detected on chromosomes 4B, 4D and 5A in the present study. The confounded QTLs on 4B, 4D and 5A had been previously detected in the Mexican conditions as deduced by their proximity to *Rht/Vrn* genes and similarity in terms of a donor parent for SB resistance allele [30,31]. The importance of genes/QTLs away from the vicinity of *Rht/Vrn* has been suggested since they might be involved in true resistance [22,30]. For chromosome 4B, out of the three detected QTLs in the WC population, the initial two (40 and 59 cM) were shared by both populations with regards to their position on CS and the farthest one (94 cM, 659.0 Mb) had been identified in the Mexican conditions [30,31]. The shared QTL at 59.0 cM for WC and 56.8 cM for KC were away from the PH QTLs and aligned at a similar 37 Mb (535.1–572.1 MB) region, being close to the newly identified *Sb4* gene (580.9–582.2 Mb) [24]. This suggests that the two distal QTLs on 4B in the WC population were significant in both the South Asian and Mexican conditions; however, the middle QTL (59 cM, 535.1–572.1 MB) representing the probable *Sb4* gene was not significant in the Mexican conditions. This implies its importance for the South Asian condition, which was detected in both populations and was different between the resistant and susceptible bulks. Although the same susceptible parent, CIANO T79, was used in both populations, not many common QTLs were identified. This must be due to the different resistance genes in the two resistant parents. Of the QTLs with CIANO T79 as the resistance donor, only the ones on 4B were shared by both populations, due to their absence in the two resistant parents. The remaining QTLs on 2A, 2B, and 4D, however, were only detected in the KC population, implying their absence in the parent KATH but presence in the parent WUYA. Such information is useful to understand the resistance gene/locus structure in the parents.

In an attempt to identify the underlying genes for the identified QTLs (Appendix A), many disease resistance-related genes within the confidence intervals of the identified QTLs were found, belonging to the ATP-binding cassette (ABC) transporter, nucleotide-binding site–leucine-rich repeat (NBS-LRR) and nucleotide binding-ARC (NB-ARC) domain gene families reported to provide durable or race specific resistance [44,45,46]. The genes have different roles in sensing pathogen effectors to trigger plant immunity [46], transportation of ions like Ca^2+^ through calmodulin-binding family protein to trigger pathogenesis-related genes [47] and triggering programmed cell death to cause leaf senescence or a hypersensitive reaction at the site of infection [48]. A higher number of QTLs in a RIL/genotype resulting in lowered SB score found in the present study have confirmed what has been suggested in previous studies [30,31]. The finding of *Lr46* giving resistance against SB in Dinajpur (1B QTL) indicates the importance of durable resistance as it is race non-specific and its effectiveness through *Lr34* (close to *Sb1* gene) and *Lr46* have been identified and discussed [19,49]. Additionally, the probable *Sb4* gene in both populations, along with identified novel QTLs, especially those differing between the bulks (2A, 2B, 2D & 5A) can also be effectively utilised for SB resistance.

## 4. Materials and Methods

### 4.1. Planting Material

Two bi-parental recombinant inbred lines (RILs) mapping populations that had been previously phenotyped and identified for SB resistance QTLs in Mexican conditions [28,29] were used in the present study. The resistant parents of the bi-parental crosses were CIMMYT breeding lines WUYA (WAXWING*2/CIRCUS) and KATH (WHEATEAR/KRONSTAD F2004) having demonstrated high resistance against the SB in previous trials and studies. They were crossed with a common susceptible parent CIANO T79 (BUCKY/(SIB)MAYA-74/4/BLUEBIRD//HD-832.5.5/OLESEN/3/CIANO-67/PENJAMO-62). The F_2_ progeny of the crosses WUYA × CIANO T79 (designated WC population) and KATH × CIANO T79 (designated KC population) were advanced through the single seed descent method to generate 231 and 230 F_2:7_ recombinant inbred lines (RILs), respectively.

### 4.2. Field Evaluation and Agronomic Practices

Field evaluation was conducted for three consecutive years 2012–2013 (designated 2013), 2013–2014 (designated 2014) and 2014–2015 (designated 2015) at Dinajpur, Bangladesh for both populations. Sowing was performed in the second fortnight of December in a randomised block design with two replications. Sowing was deliberately delayed to expose the crop to weather conditions conducive for SB development (weather data are available in Appendix A). Each genotype was sown in a two-rowed plot of 1-metre length with spacing of 20 cm between rows and a plant-to-plant distance of 5 cm. A basal dose of 66:60:60:20 kg/ha of N: P_2_O_5_: K_2_O: S from urea, triple super phosphate (TSP), muriate of potash (MoP) and gypsum along with 5 tonnes/ha of well decomposed cowdung was applied in the soil (sandy loam type) at the final land preparation stage. At the crown root initiation (CRI) stage, 34 kg/ha N from urea was top-dressed in the crop. Three irrigation treatments were performed: at 17–21 days after sowing (DAS) at CRI; 50–55 DAS at pre-heading and 70–75 DAS at the grain filling stages, with one supplementary irrigation applied whenever needed. No herbicide, insecticide or fungicide were used in crop raising and only one hand weeding at 25–30 DAS was performed. Each plot (whole plot) was harvested manually at full maturity, and threshed by an electrically operated single plot thresher.

### 4.3. Inoculation and Disease Scoring

The experimental blocks were surrounded by three spreader rows of SB susceptible cultivars (seed mixture of CIANO T79, Sonalika and Kanchan). To increase the disease pressure, the crop was inoculated with highly virulent isolates (BsDin11-1.4, BsDin11-1.1 and BsJES12-5.1) of *B. sorokiniana* at heading (Zadok’s growth stage, GS 55), with a spore suspension culture of approximately 10^4^ conidia per ml. The pure culture of *B. sorokiniana* was maintained in a potato dextrose agar (PDA) medium, which was mass multiplied on soaked and autoclaved sorghum grains. Inoculation was performed in the evening, and a light irrigation was given the next day to ensure high humidity that favours disease development. SB severity was recorded at GS 75–77, following the Saari and Prescott’s double-digit (00–99) severity scale, where the first (D1) and second (D2) digits indicate the progress of disease vertically from the ground and percentage of leaf area infected with SB, respectively. The disease severity was calculated as follows:$$\mathrm{Disease}\;\mathrm{severity}\;\left(\%\right)=\frac{D1}{9}\times \frac{D2}{9}\times 100$$

The percentage of disease severity was scored three times for 2014 and 2015 in both populations, which was used to calculate the area under the disease progress curve (AUDPC) using the formula:$$\mathrm{AUDPC}=\sum_{i=1}^{n}[\left(\frac{Y_{i}+{\;Y}_{\left(i+1\right)}}{2}\right)\times \left(t_{\left(i+1\right)}-{\;t}_{i}\right)]$$
where Y_i_ and Y_i+1_= SB percent disease severity at time t_i_ and t_(i+1)_, respectively; t_(i+1)_−t_i_ = number of days between the two disease observations; and *n* = number of times SB percent disease severity was recorded. However, SB severity was evaluated only once in 2013 for both populations and thus disease severity instead of AUDPC was used. SB severity in 2013 or AUDPC in 2014 and 2015, and their three-year averages (henceforth called SB scores) were used for QTL mapping. DH and PH (cm) was recorded in all three years for both WC and KC populations, except for PH in 2013 for KC population.

### 4.4. Statistical Analysis

Phenotypic data were subjected to statistical tests using two different statistical softwares viz. OPSTAT (http://14.139.232.166/opstat/, accessed on 25 April 2021) and Meta-R (https://data.cimmyt.org/dataset.xhtml?persistentId=hdl:11529/10201, accessed on 25 April 2021). Analysis of variance was performed using the OPSTAT software. Pearson correlation coefficients (phenotypic and genotypic) and broad-sense heritability for SB were estimated using the Meta-R software. Heritability estimates in a broad sense were calculated following the formula H2=σg 2+σg*y2y+σe2ry [50], in which σ^2^*_g_* is the variance due to genotype, σ^2^*_g_**_∗y_* is the variance due to genotype-by-year interaction, σ^2^*_e_* is the error variance, *y* is the number of years, and *r* is the number of replications. Levene’s test using ‘R’ software was carried out between two replications within the same year, and when the two replications varied significantly they were then analysed separately, otherwise they were averaged for subsequent analysis. Best linear unbiased estimate (BLUE) values for SB scores (% disease severity in 2013 and AUDPC in 2014, 2015) adjusted for heading and height were estimated using META-R software. This was meant to detect the residual effect of SB resistance for QTLs where SB resistance coincided with PH and/or DH.

### 4.5. Genotyping

Genomic DNA was isolated from the young leaves using the CTAB method. Thereafter, the DArTseq platform was used for genotyping the two populations (WC and KC) at the Genetic Analysis Service for Agriculture (SAGA) at CIMMYT, Mexico. Additional gene-based SNP markers on different chromosomes, including those for height (*Rht-B1*, *Rht-D1*), flowering (*Vrn-A1*) etc. were genotyped using the KASPar technology at CIMMYT [51]. Only high-quality markers were used for QTL analysis by removing markers with more than 20% of missing data-points, less than 30% minor allele frequency (MAF), and redundancy using the BIN function ICIMapping ver. 4.2 software [52].

### 4.6. QTL Mapping and Projection on Chinese Spring Reference Genome

The two populations were initially scored for 18,000 GBS markers out of which 2478 and 2139 non-redundant, high-quality SNP markers were filtered out for constructing a linkage map in WC and KC populations, respectively. A high-density consensus map of GBS markers made by Li et al. [53] was used to anchor the markers of the present study to their respective linkage groups. Grouping, ordering and rippling commands in the MAP function using the default parameters with a LOD value of 10 was performed in the ICIMapping ver. 4.2 software to construct the genetic map [52]. QTL mapping was performed by the BIP function of the software. QTLs were identified by both ICIM and IM methods and all the significant QTLs were reported. A QTL exceeding a threshold of LOD 3.4 for WC and 3.5 for KC (1000 permutations at alpha 0.05) in at least one environment was considered to be significant. However, if a QTL exceeded the LOD threshold of 2.5 in multiple environments, it was also considered a putative QTL. When the phenotypic variation explained (PVE) by an identified QTL is less than 10% in a single environment, it is designated as minor otherwise, a large effect QTL. A QTL with PVE larger than 10% in multiple environments is designated as a major QTL. The linkage map and LOD curves of the QTLs were drawn with software MapChart v. 2.3 [54]. The interval markers of the significant QTL were physically located on the Chinese Spring (IWGSC Chinese Spring RefSeq ver. 1.0) reference genome by BLASTN search function (expect threshold-10) using viroblast on Triticeae Toolbox (https://triticeaetoolbox.org/wheat/viroblast/viroblast.php, accessed on 25 April 2021). Candidate genes and their putative functions within the identified QTL intervals were identified using JBrowse tool for Chinese Spring (http://202.194.139.32/jbrowse-1.12.3-release/, acccessed on 25 April 2021).

## 5. Conclusions

Quantitative inheritance of SB resistance involving many QTLs having minor phenotypic effects (PVE <10%) has been reported in the past and re-established in the present investigation. It is important to note that the identified QTLs act in an additive fashion and hence stacking of the QTLs, particularly in the upcoming era of genomic selection, can be an effective breeding strategy. Many of the identified QTLs, like those on 1B, 2B, 2D, 4B, 4D and 5A were detected in both the Indian subcontinent and Mexican conditions, indicating their stability in diverse environments and the possibility of mutual detection and use. However, a few others, like those on 1A, 2A, 4D, 5D, 7B, which were found exclusively in South Asia, indicate the high G × E interaction experienced by these QTLs, which might make them highly effective in the specific environmental conditions of South Asia. The detection of resistant genes (i.e., *Lr46)* and QTLs having gene families like ABC transporter within their confidence interval indicate the importance of durable resistance against the SB. Moreover, in both populations, resistance alleles of the identified QTLs being contributed by both resistant and susceptible parents signified the dispersion of resistance genes in two parents, thus corroborating the quantitative mode of inheritance and importance of transgressive breeding to achieve SB resistance.

## Figures and Tables

**Figure 1 plants-10-00973-f001:**
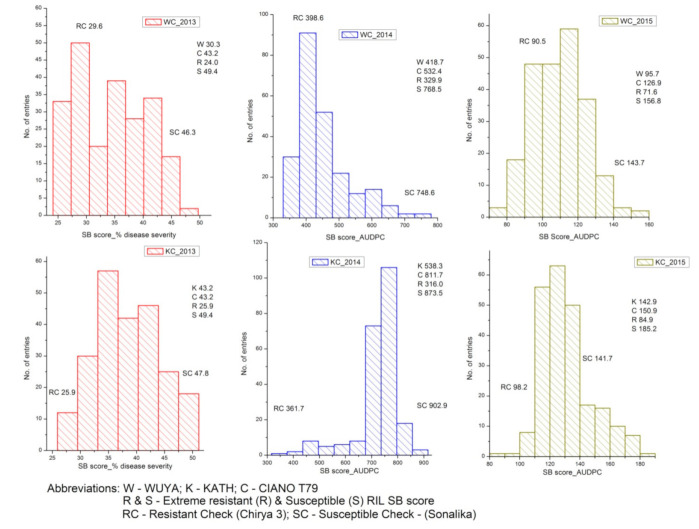
Frequency distribution of spot blotch scores (% disease severity in 2013 and AUDPC in 2014 and 2015) for three years in WC (WUYA × CIANO T79) and KC (KATH × CIANO T79) populations.

**Figure 2 plants-10-00973-f002:**
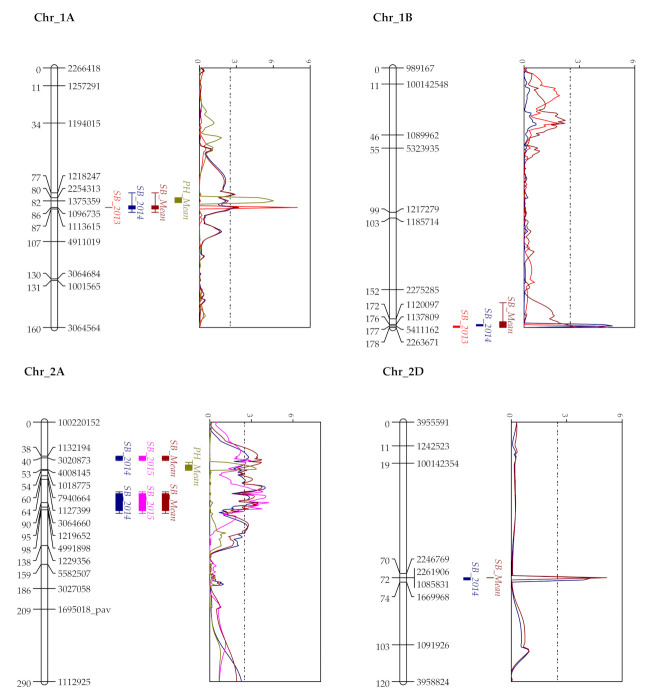
QTL profiles of SB score (% disease severity in 2013 and AUDPC in 2014 and 2015) in WC (WUYA × CIANO T79) population. QTLs for DH and PH are plotted for mean year. Associated chromosomes are represented by framework markers shown in the right side and genetic positions in centimorgan in the left side. A LOD threshold of 2.5 is depicted by the vertical dashed line on LOD graph.

**Figure 3 plants-10-00973-f003:**
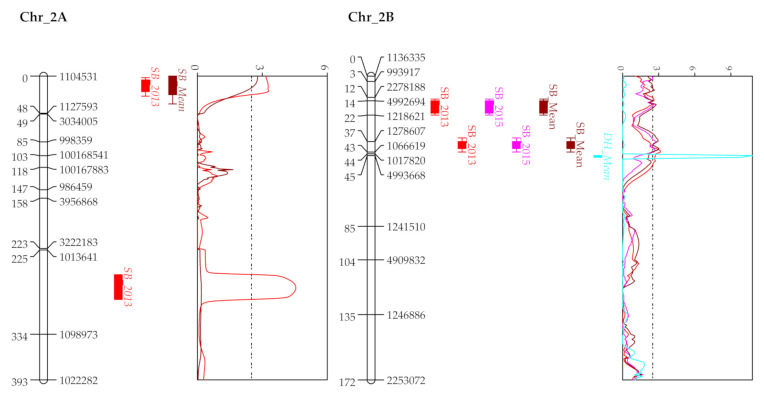
QTL profiles of SB score (% disease severity in 2013 and AUDPC in 2014 and 2015) in KC (KATH × CIANO T79) RIL population. QTLs for DH and PH are plotted for mean year. Associated chromosomes are represented by framework markers shown in the right side and genetic positions in centimorgan on the left side. A LOD threshold of 2.5 is depicted by the vertical dashed line on LOD graph.

**Figure 4 plants-10-00973-f004:**
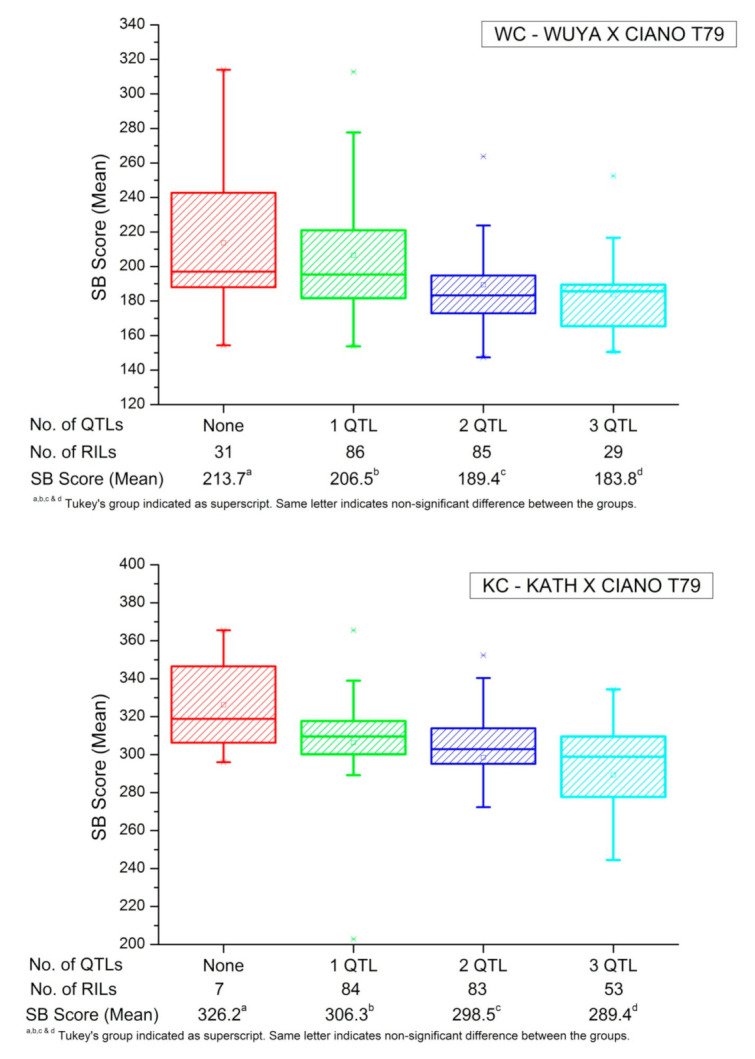
Effect of stacking three important DRSB QTLs (different between resistant & susceptible bulks) on mean spot blotch score in the two populations. For WC (2A_94.2 cM, 4B_59.0 cM, 2D_72.5 cM) and KC (2A_48.4 cM, 2B_17.9 cM, 4D_15.7 cM) populations, QTL in parentheses were considered.

**Table 1 plants-10-00973-t001:** Analysis of variance for spot blotch scores in two different RIL populations tested in Bangladesh during three crop seasons (2013–2015).

Population	Source of Variation	DF	Sum of Squares	Mean Squares	F-Calculated	Significance
WC	Genotype	230	1464.9	6.4	3.75	<0.0001
	Year	2	961.2	480.6	14.18	<0.05
	Genotype × Year	460	781.7	1.7	1.70	<0.0001
	Rep within Year	3	101.7	33.9	33.9	<0.0001
	Pooled Error	690	690.1	1.0		
	Total	1385	3999.5			
KC	Genotype	229	2453.7	10.7	2.95	<0.0001
	Year	2	39,633.4	19,816.7	275.76	<0.001
	Genotype × Year	458	1663.1	3.6	3.63	<0.0001
	Rep within Year	3	215.6	71.9	71.9	<0.0001
	Pooled Error	687	686.7	1.0		
	Total	1379	44,652.4			

**Table 2 plants-10-00973-t002:** Phenotypic and genotypic correlation of days to heading and plant height with spot blotch scores in the two RIL populations over three years along with heritability estimates for spot blotch.

Population	Years	Heading		Height		Heritability for SB
		Pcor	Gcor	Pcor	Gcor	
WC	2013	−0.271 ***	−0.306 ***	−0.561 ***	−0.678 ***	0.76
	2014	−0.282 ***	−0.340 ***	−0.471 ***	−0.586 ***	0.77
	2015	−0.206 **	−0.375 ***	−0.207 **	−0.461 ***	0.30
	Mean	−0.291 ***	−0.514 ***	−0.503 ***	−0.875 ***	0.32
KC	2013	−0.182 **	−0.234 ***	NA	NA	0.75
	2014	−0.066	−0.074	−0.214 **	−0.307 ***	0.91
	2015	−0.093	−0.095	−0.422 ***	−0.564 ***	0.68
	Mean	−0.079	−0.108	−0.328 ***	−0.624 ***	0.31

Pcor—phenotypic correlation coefficient; Gcor—genotypic correlation coefficient. Note: % disease severity in 2013 and AUDPC for 2014, 2015 and their average designated SB score were used in the analysis. ** and *** indicate significance at 1.0% and 0.1% level of probability, respectively.

**Table 3 plants-10-00973-t003:** Linkage map and SNP marker statistics for WC (WUYA × CIANO T79) and KC (KATH × CIANO T79) populations.

Chromosome	WC Population			KC Population		
	No. of Markers	Total Size (cM)	Marker Density (cM/Marker)	No. of Markers	Total Size (cM)	Marker Density (cM/Marker)
1A	148	159.8	1.1	92	88.5	1.0
1B	218	178.5	0.8	122	229.7	1.9
1D	36	116.3	3.2	34	116.9	3.4
2A	196	289.6	1.5	137	393.2	2.9
2B	175	148.4	0.8	192	172.2	0.9
2D	36	120.0	3.3	34	88.5	2.6
3A	127	247.5	1.9	169	233.2	1.4
3B	174	208.0	1.2	110	213.9	1.9
3D	17	135.5	8.0	9	166.8	18.5
4A	152	194.0	1.3	106	158.6	1.5
4B	90	180.9	2.0	50	117.6	2.4
4D	14	89.4	6.4	13	143.3	11.0
5A	173	249.1	1.4	129	228.1	1.8
5B	240	315.1	1.3	204	183.9	0.9
5D	17	96.4	5.7	16	105.6	6.6
6A	152	178.8	1.2	127	147.8	1.2
6B	207	133.5	0.6	170	96.6	0.6
6D	48	242.1	5.0	30	141.1	4.7
7A	74	149.4	2.0	166	165.8	1.0
7B	154	149.8	0.9	185	134.8	0.7
7D	30	210.3	7.0	44	191.8	4.4
Total/Average	2478	3792.4	1.5	2139	3517.9	1.6

**Table 4 plants-10-00973-t004:** QTL identified for spot blotch resistance in WC (WUYA × CIANO T79) and KC (KATH × CIANO T79) populations.

	QTL	Position (cM)	LOD	Marker Interval	Physical Range ^a^ (Mb on CS)	% PVE ^b^				R Source ^c^	Remarks ^d^
						2013	2014	2015 ^e^	Mean		
WC	1A	76.5–88.5	2.8–8.0	1218247–1096735	500.8–512.5	9.9	4.9		4.9	W	PH
	1B	175.5–178	2.8–4.8	1137809–5411162	670.6–673.7	5.4	5.4		5.1	W	*Lr46* gene
	2A	37.6–40.0	2.8–6.2	3020873–1132194	28.8–30.9		2.4	2.5 ^R1,R2^	2.6	W	PH
	2A	90.4–98.0	2.7–4.3	4991898–3064660	73.5–101.0		2.8	8.2 ^R1,R2^	3.1	W	DRSB
	2D	71.5–73.5	3.5–5.2	2261906–1085831	61.8–65.7		5.0		5.3	W	DRSB
	4B	39.5–41.5	4.8–5.7	985312–1241652	37.5–65.1	**13.7**	5.5		6.8	C	PH/*Rht-B1*
	4B	57.5–60.5	2.5–3.4	1159447–13375761	558.6–572.1	4.8		3.7 ^R1,R2^		C	DRSB/*Sb4* gene
	4B	92.5–96.5	4.1–4.4	1195526–100167524 100167524	658.0–659.5		4.5		4.9	W	
	5A	139.3–156.1	3.1–7.1	1102419–5411712	570.2–584.6	5.8	8.5		7.8	W	DH/*Vrn-A1*
	5D	38.5–51.5	7.1	1058378–1048778	458.8–542.6	**15.0**				W	
	7B	8.5–12.5	2.7–4.2	7341261–1007925	18.1–18.1	5.1	3.0		4.1	W	
					Total PVE (%)	59.7	42.0	14.4	44.6		
KC	2A	48.1–48.6	2.5–3.3	1127593–3034005	7.9–8.0	5.3			3.2	C	DRSB
	2A	224.8–334.0	4.5	1098973–1013641	16.8–65.1	7.6				K	
	2B	13.7–22.0	2.6–2.8	1218621–4992694	24.1–27.2	5.1		4.3 ^R1^	5.4	C	DRSB
	2B	37.1–43.0	2.6–3.0	1066619–1278607	56.9–454.8	6.5		4.2 ^R1,R2^	5.3	C	DH
	4B	31.8–48.0	3.1–5.7	3222467–1233562	21.6–483.8	**10.8**		**11.5 ^R1^**	7.8	C	PH/*Rht-B1*
	4B	56.5–57.0	3.8	1863994–1132777	535.1–548.2	6.6				C	*Sb4* gene
	4D	0–30.5	2.9–4.7	*Rht-D1*-BS00036421	18.8–32.3	2.3	6.5	9.1^R1^	9.3 ^R1^	K	PH/*Rht-D1/DSRB*
	4D	85.5–100.5	4.1	2257171–2256312	457.5–502.6	7.8				C	
					Total PVE (%)	52.0	6.5	29.1	31.0		

^a^ The physical position of QTL projected on IWGSC Chinese Spring RefSeq ver. 1.0 reference genome is shown in mega base pairs. QTLs with PVE >10% are in bold. ^b^ % PVE—percentage of explained phenotypic variation for a QTL. QTL exceeding threshold LOD of 3.4 for WC, 3.5 for KC in at least one environment or LOD of 2.5 in multiple environments were considered. ^c^ W: WUYA, K: KATH, C: CIANO T79. ^d^ DH: days to heading; PH: plant height; DRSB: QTL different between the resistant and susceptible bulks. ^e^ QTL analysis for individual replications was carried out in 2015 due to the significant Levene test result between replications. Superscript to a PVE in 2015 signifies whether the QTL was detected in replication 1 (^R1^), replication 2 (^R2^) or both (^R1,R2^).

**Table 5 plants-10-00973-t005:** Bulk analysis for spot blotch resistant and susceptible bulks (*n* = 10) in WC (WUYA × CIANO T79) and KC (KATH × CIANO T79) populations.

Population	Type of Bulk	SBmean ± S.E.	Average No. of QTLs in an Individual	DH Mean ± S.E.	PH Mean ± S.E.	QTLs Different between the Bulks *
WC	Resistant	156.5 ± 3.1	4.8	70.8 ± 0.6	118.8 ± 1.6	2A_94.2 (6), 4B_59.0 (6), 2D_72.5 (5)
	Susceptible	283.5 ± 7.2	1.9	66.7 ± 0.4	90.3 ± 1.5	
KC	Resistant	200.1 ± 5.9	3.0	68.7 ± 0.4	117.4 ± 1.8	2A_48.4 (4), 2B_17.9 (5), 4D_15.7 (8)
	Susceptible	351.5 ± 3.0	1.7	67.9 ± 0.5	96.8 ± 3.0	

* No. of individuals carrying the QTL out of 10 resistant RIL bulk are indicated in parentheses. DH: days to heading; PH: plant height; S.E.: standard error.

## Data Availability

The original contributions presented in the study are publicly available. These data can be found here: https://hdl.handle.net/11529/10548566.

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
