# Peer review of "Identification of QTLs for Spot Blotch Resistance in Two Bi-Parental Mapping Populations of Wheat"

_plants, 2021, doi:10.3390/plants10050973_

Round 1

Reviewer 1 Report

The manuscript describes the identification of QTLs associated with spot blotch resistance in two wheat populations. The two populations were phenotyped in natural fields with a high disease pressure. The authors identified multiple QTLs for both populations. All the methods were correct, and the results were appropriately interpreted. The manuscript is well written and organized. A few questions and concerns listed below need to be addressed by the authors.

The authors stated that all the QTL identified are minor. What is the standard or criteria for minor or major QTL?

Based on the ANOVA table, the difference between reps was significant. How did authors use the data from different reps? If the reps were not homogenous, the data should not be pooled.

A few QTL with relatively bigger PVE colocalized with agronomic traits. How confident are the authors that those QTL are real, not due to the effects from agronomic traits? How would you confirm those QTL?

Resistant alleles in WC population were mainly derived from W while resistance alleles for KC population were derived from C, the susceptible parent. What are possible reasons? The same susceptible parent was used in both populations, but there is no QTL identified in both populations? What does that mean?

The results of SB QTL mapping in Mexico using these two populations should be mentioned in detail. How are they different from this study? What are the reasons for the different results?

The authors need to justify the reason why disease severity was used in 2013 but AUPDC in 2014 and 2015. Which one would be better?

Table 2 what are Pcor and Gcor?

What is DRSB and what was the methods for this analysis?

Figure 3 QTL maps were too small to see.

Table 5 add agronomic trait values for those resistant and susceptible lines.

Line 320: should be 104  ?

Author Response

Reviewer comments: The manuscript describes the identification of QTLs associated with spot blotch resistance in two wheat populations. The two populations were phenotyped in natural fields with a high disease pressure. The authors identified multiple QTLs for both populations. All the methods were correct, and the results were appropriately interpreted. The manuscript is well written and organized. A few questions and concerns listed below need to be addressed by the authors.

Reviewer comments: The authors stated that all the QTL identified are minor. What is the standard or criteria for minor or major QTL?

RESPONSE: When the phenotypic variation explained (PVE) is less than 10% it is designated as a minor QTL. QTLs expressing PVE greater than 10% in a single environment are called large effect QTL (mentioned at Line No. 222-223, 237-238). Major QTLs are the one which expressed PVE greater than 10% in multiple environments, though present study was able to find none of such QTL. Hence, in results and discussion, we prefixed word mostly before minor QTLs as most found QTLs belonged to minor category. However, a few large effect QTLs (single environment) were also found and mentioned in Results and Discussion sections (mentioned at Line No. 222-223, 237-238, 358 etc.)

As rightly pointed out, criteria have been added in the materials and methods section 5.6 (Line No. 508 - 511)

Reviewer comments: Based on the ANOVA table, the difference between reps was significant. How did authors use the data from different reps? If the reps were not homogenous, the data should not be pooled.

RESPONSE: Replications can have significant differences in the field experiments.

1) if you check the sum of squares, the one for rep is the smallest, even smaller than that of the Error effect. The reason why it became significant was only due to its low degree of freedom leading to a bigger MS, and the high number of entries in this study leading to a smaller MS for error.

2) Before doing the analysis we have checked the phenotypic correlations between replications. They are relatively high at r (Pearson correlation coefficient) around 0.6, which is good for field experiments. Also, we did QTL analysis for individual replications and there also we obtained similar results (with similar interval markers) and more or less similar PVE values.

3) Other QTL mapping studies also had significant effects of replication, including the two papers for the same populations done in Mexico, as well as a few more papers mentioned below

He, X.; Dreisigacker, S.; Sansaloni, C.; Duveiller, E.; Singh, R.P.; Singh, P.K. Quantitative trait loci mapping for spot blotch resistance in two biparental mapping populations of bread wheat. Phytopathology 2020, 110, PHYTO-05-20-019, doi:https://doi.org/10.1094/PHYTO-05-20-0197-R.

Kumar, U., Joshi, A. K., Kumar, S., Chand, R., and Röder, M. S. 2010. Quantitative trait loci for resistance to spot blotch caused by Bipolarissorokiniana in wheat (T. aestivum L.) lines ‘Ning 8201’ and ‘Chirya 3’. Mol. Breed. 26:477-491.

Kumar, U., Joshi, A. K., Kumar, S., Chand, R., and Röder, M. S. 2009. Mapping of resistance to spot blotch disease caused by Bipolaris sorokiniana in spring wheat. Theor. Appl. Genet. 118:783-792

Singh, P.K.; He, X.; Sansaloni, C.P.; Juliana, P.; Dreisigacker, S.; Duveiller, E.; Kumar, U.; Joshi, A.K.; Singh, R.P. Resistance to spot blotch in two mapping populations of common wheat is controlled by multiple QTL of minor effects. Int. J. Mol. Sci. 2018, 19, doi:10.3390/ijms19124054.

Reviewer comments: A few QTL with relatively bigger PVE colocalized with agronomic traits. How confident are the authors that those QTL are real, not due to the effects from agronomic traits? How would you confirm those QTL?

RESPONSE: Yes, co-localized SB resistance QTLs reduced in their PVE when adjusted for days to heading and plant height (through BLUE value for SB scores adjusted for heading and height estimated through META-R software). Yet, they had residual effects (PVE) after adjustment, which indicated resistance genes are also lying in the QTL region. But, as rightly pointed out, that’s why throughout the discussion, importance has been given to QTLs without the confounding effect of heading and height.

As rightly pointed out, in results section 2.3.1 (Line No. 227-229) and 2.3.2 (Line No. 241-243) accordingly, results for reduced PVE after adjusting heading and plant height have been incorporated.

Reviewer comments: Resistant alleles in WC population were mainly derived from W while resistance alleles for KC population were derived from C, the susceptible parent. What are possible reasons? The same susceptible parent was used in both populations, but there is no QTL identified in both populations? What does that mean?

RESPONSE: As correctly observed, the resistant alleles from the parent CIANO T79 (C) were found more in KC population than WC population in the present study. However, CIANO T79 (C) has provided SB resistant alleles in both WC and KC population. In fact, CIANO T79 (C) has provided resistant alleles in the Mexico conditions on chromosome 1B, 1D, 2B, 3A, 4B in four different RIL populations (including WC and KC); a few of whom (1D, 2B, 4B) had been reconfirmed in South Asia conditions as well and discussed in the present study. Higher number of QTLs having ‘C’ alleles as the SB resistant donors in KC population (compared to WC) indicates the difference between the two resistant donors i.e., WUYA (W) and KATH (K) especially in Indian subcontinent conditions where KATH performed poorly compared to WUYA in SB disease score. Particularly in 2013 and 2015 KATH (K) and CIANO T79 (C) scored comparable in SB disease score (Indicated in Line No. 101-102). This shows that the resistance genes do not concentrate in any one parent but get accumulated from both the parents, which yet substantiates the quantitative inheritance for SB resistance.

Rightly pointed out addition has been made in the conclusion section (Line No. 413-417)

Common QTLs have been identified in both population (Line No. 320-322). From CIANO T79 perspective as well (C as resistant donor for SB resistance), a common QTL site on 4B (around 40 cM in both WC and KC) which co-localized with plant height (Rht-B1) and was also similarly present in Mexico conditions (WC population), was found in the present study, which has been discussed at Line No. 376 – 385.

Reviewer comments: The results of SB QTL mapping in Mexico using these two populations should be mentioned in detail. How are they different from this study? What are the reasons for the different results?

Response: Reference No. 30 and 31 belong to SB QTL mapping in Mexico conditions. QTLs in Mexico conditions have been discussed throughout the discussion section and compared thereof with the results of the present study, e.g., for 1B QTL (Line No. 336-338 ), 2B, 2D QTLs (Line No. 348-352), 4B, 4D, 5A (Line No. 368-371). The similarity and differences in the obtained results for QTLs (Mexico vs South Asia) were also discussed in the lines mentioned above in the discussion section.

Further additions have been made in the conclusion section (Section 4, Line No. 406-411)

Reviewer comments: The authors need to justify the reason why disease severity was used in 2013 but AUPDC in 2014 and 2015. Which one would be better?

RESPONSE: Disease severity at three subsequent intervals (post flowering, weekly interval) was used to calculate AUDPC values which is surely more informative than disease severity alone. Due to some unavoidable circumstances, AUDPC was not recorded in 2013, though the recording is made post-heading at a much latter stage (equivalent to third/final disease severity observation of 2014 and 2015), so that there was full expression of the disease. The 2013 observations are substantiated in identified QTLs (Table 4) where few QTLs identified in 2013, reappeared in 2014, 2015 and mean year with similar QTL intervals and donors of the resistant alleles. That is why they were included in the study.

Reviewer comments: Table 2 what are Pcor and Gcor?

RESPONSE: Pcor is phenotypic correlations (Pearson correlation value using phenotypic values). Gcor is Genotypic correlation (Pearson correlation coefficient values after taking out the environmental/error variances). Both are calculated by the Meta-R software detailed in section 4.2.

As rightly pointed out, abbreviations are added in the Table 2 (Line No. 115)

Reviewer comments: What is DRSB and what was the methods for this analysis?

RESPONSE: The QTLs which majorly varied between the resistant and susceptible bulks were named DRSB (different between resistant and susceptible bulks). The method is detailed in section 2.3.3 with additions made at Line No. 303-304. Left and right interval markers of each of identified QTLs is scored on each of the RIL of the bulk (resistant or susceptible). When favourable (resistant) allele is present for both left and right marker of the QTL interval, then only QTL is marked to be present.

e.g.

QTL on 1A in WC population is present in 40% RILs (4 out of 10 RILs) of the resistant bulk. Same QTL in susceptible bulk is present is 30% RILs (3 out of 10 RILs). Thus, 4 – 3 = 1, it is not much differing QTL between the two bulks.

QTL on 2D in WC population is present in 50% RILs (5 out of 10 RILs) of the resistant bulk. However, this particular QTL is totally absent in susceptible bulk (0 out of 10 RILs) in WC population. Thus, 5 – 0 = 5, it is the differing QTL between the two bulks.

Three top differing QTLs for WC and KC populations are accounted in the table 5.

Reviewer comments: Figure 3 QTL maps were too small to see.

RESPONSE: Size and resolution of QTL maps has been improved.

Reviewer comments: Table 5 add agronomic trait values for those resistant and susceptible lines.

RESPONSE: Days to heading and plant height data has been added in Table 5.

Reviewer comments: Line 320: should be 104  ?

RESPONSE: Corrected and thanks

Reviewer 2 Report

Dear authors,

this manuscript is very interesting considering spot blot disease in some parts of the world, due to yield losses up to 85% in Zambia, for example. In introduction you could also mention about fungicide protection, or better to say how were the yield losses with the usage of fungicides? What is the active components most efficient against SB? In material and methods please describe more about agronomical practice used in field trials (herbicides, insecticides, I assume that fungicides were omitted, fertilization, soil type, weather conditions and so on. For example you can make climatic diagrams as you have three years trials. That would be interesting to see in comparison to disease scooring. Furthermore in materials and methods, you could make different subtitles. For example: seperate Inoculum production and disease scooring against Plant material.

Once more check the whole manuscript; Here are just few suggestions:

in abstract use italic for genes (Sb4, Lr46...). 

Pg2 line 55 It is more difficult to apply fungicide 

Add reference for investigation about issue with fungicides

Pg10, line 204 temperatures and rainfall

                       during the grain filling stage

Author Response

Reviewer comments: This manuscript is very interesting considering spot blot disease in some parts of the world, due to yield losses up to 85% in Zambia, for example.

In introduction you could also mention about fungicide protection, or better to say how were the yield losses with the usage of fungicides? What is the active components most efficient against SB?

RESPONSE: Lines have been added in the Introduction section (Line No. 50 to 58)

Both seed treatments and foliar fungicidal sprays are advocated against SB disease. Seed treatment with a mixture of carboxin and thiram or fludioxonil and difenoconazole not only limits the disease severity but also improves seed germination in fields [13,14]. Likewise foliar sprays with a number of fungicidal molecules like carbendazim, difenoconazole, propiconazole, azoxistrobin have been tested against SB infection and found to reduce yield reduction to the tune of 10 – 30%. The high efficacy of triazole fungicides like tebuconazole and propiconazole is attributed to their ability to limit SB infection by interfering with fungal cell wall synthesis [14,15]. Despite the proven efficacy of fungicides, they have not been widely applied due to high costs, limited availability, and hazard to the environment. This is especially true for South Asian countries, where 80-90% of farmers are small-holders [3,9].

Reviewer comments: In material and methods please describe more about agronomical practice used in field trials (herbicides, insecticides, I assume that fungicides were omitted, fertilization, soil type, weather conditions and so on.

RESPONSE: Agronomic practices added (Line No. 435 to 445) in section 5.2

5.2 Field evaluation and agronomic practices

Sowing was deliberately delayed to expose the crop to weather conditions conducive for SB development (Weather data are available in Supplementary Figure S2).Each genotype was sown in a 2-rowed plot of one-meter length with a spacing of 20 cm between rows and a plant-to-plant distance of 5 cm. A basal dose of 66:60:60:20 kg/ha of N: P2O5: K2O: S from urea, triple super phosphate (TSP), Muriate of potash (MoP) and gypsum along with 5 tonnes/ha well decomposed cow-dung was applied in the soil (sandy loam type) at final land preparation stage. At crown root initiation (CRI) stage, 34 kg/ha N from urea was top-dressed in the crop. Three irrigation at 17 – 21 days after sowing (DAS) at CRI, 50-55 DAS at pre-heading and 70-75 DAS at grain filling stages, with one supplementary irrigation applied whenever needed. No herbicide, insecticide or fungicide were used in crop raising and only one hand weeding at 25-30 DAS was performed. Each plot (whole plot) was harvested manually at full maturity, and threshed by an electrically operated single plot thresher.

Reviewer comments: For example you can make climatic diagrams as you have three years trials. That would be interesting to see in comparison to disease scooring.

RESPONSE: A supplementary diagram has been made. (Supplementary Figure S2)

Reviewer comments: Furthermore in materials and methods, you could make different subtitles. For example: seperate Inoculum production and disease scoring against Plant material.

RESPONSE: Separate heading have been made in Materials and methods section

Section 5.1 Planting material

5.2 Field evaluation and agronomic practices

5.3 Inoculation and disease scoring

5.4 Statistical analysis

5.5 Genotyping

5.6 QTL mapping and projection on Chinese spring reference genome

Reviewer comments: Once more check the whole manuscript; Here are just few suggestions:

in abstract use italic for genes (Sb4, Lr46...). 

RESPONSE: Thanks and Corrected (Line 24 and 25)

Pg2 line 55 It is more difficult to apply fungicide 

Add reference for investigation about issue with fungicides

RESPONSE: Detailed aspect on fungicides along with references have been added (Line No. 50 – 58)

Pg10, line 204 temperatures and rainfall during the grain filling stage

RESPONSE: Line No. 310, references are there.

Reviewer 3 Report

Please see attached for comments. 

Author Response

Reviewer comments: Please make the expression consistent with reference to below

RESPONSE: Thanks and Done (Line 33 and 64)

Please further explain why you think your research/materials were distinct from previous researches, as there are tons of similar researches focusing on SB in wheat?

RESPONSE: Additions have been made in Line No. 81 to 85 and 87.

Reviewer comments: Suggest to include a short and reasonable discussion on why D genome has lowest marker density as compared to A and B.

RESPONSE: The issue has been discussed in the Discussion section (Line No. 322 – 325) along with the references.

Although lower genetic coverage was observed for D genome compared to A and B genomes, present study was able to locate SB resistance QTLs on 2D, 4D and 5D. Many past studies have also indicated low polymorphism in D genome, which may be attributed to its recent introgression into wheat genome [37,38].

Reviewer comments: Very good point as a starting point for further uncovering its resistance mechanism. I would like to see the results of predicated potential candidate genes within the most (if all cannot be fulfilled) QTL, followed by a discussion. Some promising candidate genes may or may not be identified, as Chinese Spring may or may not harbor the resistance genes/QTL.

RESPONSE: Two supplementary tables (Supplementary table S3 and S4) have been added in the manuscript. Important findings have been discussed in the Discussion section (Line No. 386-390). Changes have been made in Materials and method section as well (Line No. 515-517)

In an attempt to in-silico identify mode of action of the identified QTLs (Supplementary table S3 and S4), there are many disease resistance-related genes within the confidence intervals ofthe identified QTLs,belonging to ATP-binding cassette (ABC) transporter, nucleotide-binding site–leucine-rich repeat (NBS-LRR) and nucleotide binding – ARC (NB-ARC) domain gene families reported to provide durable or race specific resistance [44-46]. 

Round 2

Reviewer 1 Report

The authors responded to most concerns and questions and made corresponding changes. There are still a few that the authors did not completely addressed.

Homogeneity test should be done for different reps in order to see if they can be combined.

The authors did not make a complete response to comments about same and different QTL identified in two population which had the same susceptible parent. please indicate in discussion what does that mean when different QTL were identified and what does that mean when same QTL were identified.

Please make note in the manuscript on the reason why disease severity was used in 2013

Line 414, it is not confirmed if the QTL is Sb4. Thus, you cannot say the detected sb4.

Fig. 3 is still too small to see.

Author Response

The authors responded to most concerns and questions and made corresponding changes. There are still a few that the authors did not completely addressed.

Reviewer’s comments: Homogeneity test should be done for different reps in order to see if they can be combined.

Response: Levene’s test for homogeneity of variances was carried out between the two replications within a same year for both populations in all three years. As indicated in the revised manuscript, the result was non-significant for both populations in 2013 and 2014, but was significant in 2015, thus the two replications of both populations in 2015 were analyzed separately for QTL mapping as indicated in Table 4. Accordingly new QTL profiles have been updated. It is noteworthy that we’ve identified a mistake in calculating AUDPC for the 2015 experiments, when we were re-checking the original datasets for Levene’s test. We’ve re-calculated the 2015 and mean data and updated the relevant tables and figures. This revision, however, didn’t change the general trends in the results, and most QTLs in the year 2015 remained significant, with only slightly changed PVE values and positions.

Reviewer’s comments: The authors did not make a complete response to comments about same and different QTL identified in two population which had the same susceptible parent. please indicate in discussion what does that mean when different QTL were identified and what does that mean when same QTL were identified.

Response: A paragraph has been added in discussion to address this issue.

Reviewer’s comments: Please make note in the manuscript on the reason why disease severity was used in 2013

Response: Suggestion accepted and the information was added in the Materials and Methods section.

Reviewer’s comments: Line 414, it is not confirmed if the QTL is Sb4. Thus, you cannot say the detected sb4.

Respone: Agreed. It is probably Sb4. We’ve made revision accordingly in the manuscript.

Reviewer’s comments: Fig. 3 is still too small to see.

Respone: Both Fig 2 and Fig 3 are newly made. Framework markers have been reduced and font sizes have been increased to enhance visibility.